# Toll-like receptors 1–9 in small bowel neuroendocrine tumors–Clinical significance and prognosis

Niko Hiltunen[1]*, Niko Kemi[1], Juha P. Väyrynen[1], Jan Böhm[2], Joonas H. Kauppila[3,4], Heikki Huhta[3], Olli Helminen[1,3]

1 Translational Medicine Research Unit, Medical Research Center, University of Oulu and Oulu University Hospital, Oulu, Finland, 2 Department of Pathology, Central Finland Central Hospital, Jyväskylä, Finland, 3 Surgery Research Unit, Medical Research Center, University of Oulu and Oulu University Hospital, Oulu, Finland, 4 Department of Molecular Medicine and Surgery, Karolinska Institutet and Karolinska University Hospital, Upper Gastrointestinal Surgery, Stockholm, Sweden

* niko.hiltunen@oulu.fi

**Data Availability Statement:** All relevant data (anonymized) are within the manuscript and its Supporting Information files.

## Abstract

Toll-like receptors (TLRs) are pattern recognition receptors of the innate immunity. TLRs are known to mediate both antitumor effects and tumorigenesis. TLRs are abundant in many cancers, but their expression in small bowel neuroendocrine tumors (SB-NETs) is unknown. We aimed to characterize the expression of TLRs 1–9 in SB-NETs and lymph node metastases and evaluate their prognostic relevance. The present study included 125 patients with SB-NETs, of whom 95 had lymph node metastases, from two Finnish hospitals. Tissue samples were stained immunohistochemically for TLR expression, assessed based on cytoplasmic and nucleic staining intensity and percentage of positively stained cells. Statistical methods for survival analysis included Kaplan-Meier method and Cox regression adjusted for confounding factors. Disease-specific survival (DSS) was the primary outcome. TLRs 1–2 and 4–9 were expressed in SB-NETs and lymph node metastases. TLR3 showed no positive staining. In primary SB-NETs, TLRs 1–9 were not associated with survival. For lymph node metastases, high cytoplasmic TLR7 intensity associated with worse DSS compared to low cytoplasmic intensity (26.4% vs. 84.9%, p = 0.028). Adjusted mortality hazard (HR) was 3.90 (95% CI 1.07–14.3). The expression of TLRs 1–6 and 8–9 in lymph node metastases were not associated with survival. SB-NETs and their lymph node metastases express cytoplasmic TLR 1–2 and 4–9 and nucleic TLR5. High TLR7 expression in SB-NET lymph node metastases was associated with worse prognosis. The current research has future perspective, as it can help create base for clinical drug trials to target specific TLRs with agonists or antagonists to treat neuroendocrine tumors.

## Introduction

Neuroendocrine tumors (NETs) can originate in almost any location in the body that contains neuroendocrine cells [1]. The incidence of NETs has increased steadily over the past decades which suggests that the incidence may not have yet reached its peak [2,3]. The incidence of

**Funding:** This study was funded by The Finnish Medical Foundation grant no. 5954 (NH), Mary and Georg C. Ehrnrooth Foundation (NH, OH), Instrumentarium Science Foundation (OH) and Finnish State Research Funding (OH). The funders had no role in study design, data collection and analysis, decision to publish, or preparation of the manuscript.

**Competing interests:** The authors have declared that no competing interests exist.

NETs has increased 6.4-fold between 1973 and 2012, according to the SEER database [3]. Small bowel neuroendocrine tumors (SB-NETs) comprise approximately 13–28% of the gastroenteropancreatic NETs. Jejunum and ileum are the most common sites for SB-NETs [4,5]. NETs differ in behavior and especially patient survival depending on multiple factors such as tumor location, histology, stage, age, and sex. The 5-year overall survival for grade 1 and 2 SB-NETs is 54–71% depending on disease stage whereas 10-year survival is 30–49%. The median survival time is 115 months for jejunal/ileal disease [6].

Toll-like receptors (TLRs) are important mediators of the innate immune system. TLRs are pattern recognition receptors (PRR) that distinguish between self and non-self by identifying pathogen associated molecular patterns (PAMPs) and danger associated molecular patterns (DAMPs). PAMPS are exogenous ligands of microbial components whereas DAMPs are endogenous ligands released into the circulation in cellular stress or necrosis [7,8]. TLRs are type 1 transmembrane proteins consisting of unique cytoplasmic and extracellular antigen-recognition domains [8]. The recognition of PAMPs and DAMPs by TLRs launches a signaling cascade that ultimately leads to the activation of several transcription factors and the production of cytokines and chemokines as an immunologic response [9]. TLR1 recognizes bacterial lipoproteins in an interaction with TLR2 [10]. TLR2 recognizes a variety of microbial components including lipoproteins, peptidoglycan and lipoteichoic acid [9,11]. TLR3 recognizes double stranded viral RNA [12]. TLR4 recognizes lipopolysaccharides and several other ligands such as taxol and endogenous ligands like heat shock proteins [7,13]. TLR5 responds to flagellin, a structural protein component of the flagella of Gram-negative bacteria [14]. TLR6 recognizes di-acyl lipopeptides and has also been shown to functionally cooperate with TLR2 to differentiate between di- and tri-acyl lipopeptides [7,15]. TLRs 7 and 8 both recognize single stranded RNA [9,16]. TLR9 is responsible for directly recognizing bacterial DNA with unmethylated cytosine-phosphate-guanine motifs [17]. In humans, TLR10 is the latest discovered TLR and its ligands and functions remain unclear [18,19]. TLRs 1, 2, 4, 5, and 6 detect extracellular ligands and TLRs 3, 7, 8, and 9 detect intracellular ligands [9].

There has been growing interest around the expression of TLRs in cancer over the last years. Chronic inflammation plays a notable role in the pathogenesis of various gastrointestinal tract cancers such as gastric cancer and hepatocellular carcinoma [20,21]. TLR activation has been shown to upregulate the production of proangiogenic factors, promote invasiveness and cancer cell adhesion, and amplify tumor metastasis [22]. Some studies on the other hand have shown TLR activation to have antitumoral effect by improving the antitumoral immune response, acting on immune cells or directly on tumor cells [23]. The expression of TLRs in SB-NETs is completely unknown.

The aim of the present study was to characterize the expression of TLRs 1–9 in SB-NETs and lymph node metastases and investigate the correlation of expression with survival and clinicopathological variables in a large retrospective consecutive series of SB-NETs from two Finnish institutions with long follow-up and complete survival data. In this study, expression of TLRs 1–9 in SB-NETs was characterized for the first time.

## Materials and methods

### Patients and data collection

The initial cohort and data collection have been previously described [24]. The original data was collected between February 1, 2019 and May 1, 2019. The data was updated regarding follow-up information and survival data on April 19, 2021. The statistical analyses were performed between July 27, 2023 and August 5, 2023. This study included 125 patients who had been diagnosed and treated in Oulu University Hospital (n = 98) from February 9, 2000

until February 7, 2018 and Central Finland Central Hospital (n = 27) from February 24, 2000 until December 31, 2017. At the time of diagnosis, 76% (n = 95) of these patients presented with lymph node metastases and samples from these were also included. Clinical data were collected from electronic health record by NH and OH. The compiled dataset was anonymized. Survival data were provided by The Cause of Death Registry maintained by Statistics Finland. The end of follow-up was December 31, 2019. The study and use of samples and data was approved by the Oulu University Ethics Committee (EETTMK 81/2008). The National Authority for Medicolegal Affairs (VALVIRA) waived the need for written informed consent due to the retrospective nature of the study and approved for the use of data and samples.

Experienced pathologists determined the histological diagnoses at the time of treatment. Tumor stage was determined according to the 8th edition of the AJCC/UICC Tumor, Node, Metastasis (TNM) categories [25]. Tumor grade was determined according to the WHO 2019 classification of tumors of the digestive system [26].

## Immunohistochemical staining

Routine diagnostic tissue samples from both SB-NETs and lymph node metastases were fixed in formalin and embedded in paraffin. Representative samples including deepest tumor invasion were identified based on diagnostic hematoxylin-eosin slides. For immunohistochemistry, the tissue sample blocks were fetched from archives of the Department of Pathology of Oulu University Hospital and Central Finland Central Hospital and cut in tissue sections of 3.5 μm in thickness. The sections were deparaffinized in xylene and rehydrated through graded alcohols.

Antigen retrieval was performed with tris-EDTA buffer at pH 9 in a microwave oven first at 800W for 2 minutes and then at 150W for 15 minutes. Tissue sections were first cooled at room temperature for 20 minutes, then rinsed in distilled water and in phosphate-buffered saline containing Tween (PBS-Tween). Endogenous peroxidase activity was neutralized in peroxidase blocking solution (Dako, Glostrup, Denmark, S2023) for 5 minutes, and the tissue sections were washed in PBS-Tween for two 5-minute cycles. After this, sections were incubated with rabbit polyclonal antibodies (TLRs 1, 2, 6, 7) and mouse monoclonal antibodies (TLRs 3, 4, 5, 8, 9) in dilute solution (Dako S2022); TLR1 for 60 minutes (diluted 1:300, Abcam ab189337), TLR2 for 60 minutes (diluted 1:500, Rockland 600-401-956), TLR3 for 120 minutes (diluted 1:30, Novus NBP2-24875), TLR4 for 60 minutes (diluted 1:1000, Abnova H00007099-M02), TLR5 for overnight in +4˚C (diluted 1:75, Novus NBP2-24787), TLR6 for 60 minutes (diluted 1:750, Abnova PAB 3555), TLR7 for 60 minutes (diluted 1:500, Novus NB100-56682), TLR8 for 60 minutes (diluted 1:850, Novus NBP2-24917) and TLR9 for 60 minutes (diluted 1:300, Novus NBP2-24729). After incubation, wash in PBS-Tween was repeated twice for 5 minutes. The sections were then incubated with En-Vision polymer (Dako K5007) for 30 minutes and again washed in PBS-Tween for two cycles of 5 minutes. After the final wash, diaminobenzidine working solution (Dako K5007) was used as a chromogen. Lastly, the samples were rinsed in distilled water and counterstained in hematoxylin for 1 minute. Staining was done with Dako Autostainer (Dako, Copenhagen, Denmark). Cancer tissues with high expression of TLR were used as external positive controls. To confirm the antigen preservation in the old paraffin blocks we compared the staining intensities in SB-NETs between old and new blocks divided by the median age of the blocks. No significant differences were found.

## Immunostaining assessment

Microscopic slides were scanned at x20 magnification using an Aperio AT2 digital slide scanner. Immunoreactivity of TLRs 1–9 was independently evaluated by two researchers (NH and

NK) using QuPath [27]. We assessed cytoplasmic and nucleic staining intensity on a scale from zero to three (0–3), zero meaning no staining, one meaning weak staining, two meaning intermediate staining and three meaning strongest intensity of staining. We also assessed the percentage of positively stained cells (0–100%). All samples with intensity ≥ 1 presented with 100% positive tumor cells so only the intensity variable was used for statistical analyses. Mean values of two independent evaluations were used. Dichotomization into groups of low and high intensity was done for all TLR intensity variables. Medians were chosen as cut-off values.

## Statistical analysis

The statistical analyses were computed with IBM SPSS statistics 28 for Windows (IBM Corporation, Armonk, NY, USA). The baseline values were compared with chi square test for categorized and with Mann-Whitney U test for continuous variables. Survival time was from the date of surgery until either time of death or end of follow-up, whichever came first. The Kaplan-Meier method with log-rank test stratified by TLR intensity (low or high) was used to calculate disease specific survival (DSS). Cox regression was used to calculate crude and adjusted hazard ratios (HR). Cox regression was adjusted for age, sex, stage (I-II, III, IV), grade (G1, G2) and adjuvant somatostatin therapy (no, yes). P-values of less than 0.05 were considered significant.

## Results

### Patients

The number of adequate samples and representative immunostainings varied between different TLR stainings. Of the total cohort of 125 SB-NETs, analyzable samples were available for 116 (TLR1), 84 (TLR2), 116 (TLR4), 99 (TLR5), 115 (TLR6), 111 (TLR7), 113 (TLR8) and 111 (TLR9) cases. Of the 95 lymph node metastases, analyzable samples were available for 70 (TLR1), 72 (TLR2), 77 (TLR4), 58 (TLR5), 76 (TLR6), 72 (TLR7), 76 (TLR8) and 70 (TLR9) cases.

In the total cohort of 125 patients, 57 (45.6%) were women and 68 (54.4%) were men, the median age being 66 years (IQR 55–73). Median follow-up time was 70 months (IQR 40–125, Range 229). Of the tumors, 16 (12.9%) were stage I-II, 63 (50.8%) were stage III and 45 (36.3%) were stage IV. Forty-four deaths occurred during follow-up and 26 of these deaths were disease-specific. Overall survival during the whole follow up was 41.0%. Baseline characteristics are presented separately and divided between low and high intensity groups for TLR1 (Table 1), TLR2 (Table 2), TLR4 (Table 3), TLR5 (Tables 4 and 5), TLR6 (Table 6), TLR7 (Table 7), TLR8 (Table 8), and TLR9 (Table 9).

### Staining

Cytoplasmic staining was observed with TLRs 1–2 and 4–9. Nucleic staining was observed with TLR5. Staining examples are shown in Fig 1. The only TLR showing completely negative staining despite external control being positive was TLR3. All stained cases were evaluated to have 100% staining. Median staining intensities are shown in Table 10.

### Toll-like receptors and association with clinicopathological variables

There was a significant difference in median age of the patients between low and high TLR4 cytoplasmic intensity in SB-NETs, patients being older in the latter group (58 years vs. 68 years, p = 0.002) (Table 3). Similar finding was also true with SB-NET cytoplasmic TLR9 intensity (61 years vs. 69 years, p = 0.012) (Table 9). A larger proportion of jejunal tumors was observed in SB-NETs with high nucleic TLR5 intensity group compared to low nucleic intensity (10.8% vs. 1.6%, p = 0.043) (Table 5). Similar finding was also true with TLR7 in SB-NETs

**Table 1. Baseline characteristics and comparison stratified by Toll-like receptor (TLR) 1 cytoplasmic intensity in primary tumors and lymph node metastases.**

| Variables | TLR1 primary tumor cytoplasmic intensity low, *n = 71* | TLR1 primary tumor cytoplasmic intensity high, *n = 45* | P between groups | TLR1 metastasis cytoplasmic intensity low, *n = 61* | TLR1 metastasis cytoplasmic intensity high, *n = 9* | P between groups |
|---|---|---|---|---|---|---|
| **Sex** | | | 0.217 | | | 0.588 |
| Male, *n* (%) | 43 (60.6) | 22 (48.9) | | 33 (54.1) | 4 (44.4) | |
| Female, n (%) | 28 (39.4) | 23 (51.1) | | 28 (45.9) | 5 (55.6) | |
| **Age, median (IQR) years** | 62.026 (54.084–72.17) | 67.55 (57.34–74.78) | 0.169 | 65.64 (55.66–71.30) | 69.43 (60.21–73.33) | 0.376 |
| **T-Class** | | | 0.399 | | | 0.962 |
| T1–2, *n* (%) | 22 (31.0) | 9 (20.0) | | 13 (21.3) | 2 (22.2) | |
| T3, *n* (%) | 28 (39.4) | 22 (48.9) | | 30 (49.2) | 4 (44.4) | |
| T4, *n* (%) | 21 (29.6) | 14 (31.1) | | 18 (29.5) | 3 (33.3) | |
| **N-Class** | | | 0.538 | | | |
| N0, *n* (%) | 16 (22.5) | 8 (17.8) | | | | |
| N1–2, *n* (%) | 55 (77.5) | 37 (82.2) | | | | |
| **M-Class** | | | 0.089 | | | 0.064 |
| M0, *n* (%) | 41 (57.7) | 33 (73.3) | | 40 (65.6) | 3 (33.3) | |
| M1, *n* (%) | 30 (42.3) | 12 (26.7) | | 21 (34.4) | 6 (66.7) | |
| **Stage** | | | 0.176 | | | 0.064 |
| I–II, *n* (%) | 9 (12.7) | 5 (11.1) | | 0 | 0 | |
| III, *n* (%) | 32 (45.1) | 28 (62.2) | | 40 (65.6) | 3 (33.3) | |
| IV, *n* (%) | 30 (42.3) | 12 (26.7) | | 21 (34.4) | 6 (66.7) | |
| **Grade** | | | 0.551 | | | 0.771 |
| 1, *n* (%) | 53 (80.3) | 34 (75.6) | | 42 (70.0) | 6 (75.0) | |
| 2, *n* (%) | 13 (19.7) | 11 (24.4) | | 18 (30.0) | 2 (25.0) | |
| **Tumor location** | | | 0.567 | | | 0.496 |
| Jejunum, *n* (%) | 5 (7.0) | 2 (4.4) | | 3 (4.9) | 0 | |
| Ileum, *n* (%) | 66 (93.0) | 43 (95.6) | | 58 (95.1) | 9 (100) | |
| **Somatostatin analogue treatment** | | | 0.395 | | | 0.597 |
| Yes, *n* (%) | 42 (59.2) | 23 (51.1) | | 35 (57.4) | 6 (66.7) | |
| **Chemotherapy** | | | 0.594 | | | 0.197 |
| No, *n* (%) | 60 (84.5) | 38 (84.4) | | 52 (85.2) | 6 (66.7) | |
| Preoperative, *n* (%) | 4 (5.6) | 1 (2.2) | | 1 (1.6) | 1 (11.1) | |
| Postoperative, *n* (%) | 7 (9.9) | 6 (13.3) | | 2 (22.2) | 2 (22.2) | |
| **Multiple primary tumors** | | | 0.404 | | | 0.702 |
| Yes, *n* (%) | 18 (26.1) | 15 (33.3) | | 17 (28.3) | 2 (22.2) | |
| **P-CgA** | | | | | | |
| Median (IQR) nmol/L | 7.15 (4.00–17.00) | 5.6 (3.05–11.90) | 0.083 | 5.35 (3.30–12.00) | 5.60 (4.80–19.00) | 0.422 |
| **dU-5-HIAA** | | | | | | |
| Median (IQR) µmol/L | 54.5 (25.75–176.25) | 84.50 (27.25–218.50) | 0.620 | 77.00 (38.00–169.00) | 56.00 (38.75–168.25) | 0.902 |

P-CgA, plasma chromogranin A; dU-5-HIAA, 24 h urine 5-hydroxyindoleacetic acid; Classes T-, N- and M refer to Tumor, Node and Metastasis of the TNM-staging system.

(13.5% vs 2.7%, p = 0.027) (Table 7). A larger proportion of patients with high cytoplasmic TLR7 intensity in lymph node metastases tended to have high T-class (T3-4) compared to the low intensity group (p = 0.049) (Table 7). Patients with high cytoplasmic TLR8 intensity in lymph node metastases had received more somatostatin analogue treatment compared to low

**Table 2. Baseline characteristics and comparison stratified by Toll-like receptor (TLR) 2 cytoplasmic intensity in primary tumors and lymph node metastases.**

| Variables | TLR2 primary tumor cytoplasmic intensity low, *n = 52* | TLR2 primary tumor cytoplasmic intensity high, *n = 32* | P between groups | TLR2 metastasis cytoplasmic intensity low, *n = 37* | TLR2 metastasis cytoplasmic intensity high, *n = 35* | P between groups |
|---|---|---|---|---|---|---|
| **Sex** | | | 0.091 | | | 0.824 |
| Male, *n* (%) | 31 (59.6) | 13 (40.6) | | 20 (54.1) | 18 (51.4) | |
| Female, n (%) | 21 (40.4) | 19 (59.4) | | 17 (45.9) | 17 (48.6) | |
| **Age, median (IQR) years** | 65.55 (54.13–72.50) | 65.80 (57.28–73.25) | 0.665 | 64.83 (54.70–72.70) | 67.55 (56.27–73.71) | 0.414 |
| **T-Class** | | | 0.413 | | | 0.552 |
| T1–2, *n* (%) | 14 (26.9) | 10 (31.3) | | 7 (18.9) | 10 (28.6) | |
| T3, *n* (%) | 22 (42.3) | 9 (28.1) | | 20 (54.1) | 15 (42.9) | |
| T4, *n* (%) | 16 (30.8) | 13 (40.6) | | 10 (27.0) | 10 (28.6) | |
| **N-Class** | | | 0.770 | | | |
| N0, *n* (%) | 10 (19.2) | 7 (21.9) | | | | |
| N1–2, *n* (%) | 42 (80.8) | 25 (78.1) | | | | |
| **M-Class** | | | 0.511 | | | 0.951 |
| M0, *n* (%) | 33 (63.5) | 18 (56.3) | | 23 (62.2) | 22 (62.9) | |
| M1, *n* (%) | 19 (36.5) | 14 (43.8) | | 14 (37.8) | 13 (37.1) | |
| **Stage** | | | 0.799 | | | 0.951 |
| I–II, *n* (%) | 6 (11.5) | 3 (9.4) | | 0 | 0 | |
| III, *n* (%) | 27 (51.9) | 15 (46.9) | | 23 (62.2) | 22 (62.9) | |
| IV, *n* (%) | 19 (36.5) | 14 (43.8) | | 14 (37.8) | 13 (37.1) | |
| **Grade** | | | 0.248 | | | 0.151 |
| 1, *n* (%) | 38 (76.0) | 26 (86.7) | | 23 (63.9) | 27 (79.4) | |
| 2, *n* (%) | 12 (24.0) | 4 (13.3) | | 13 (36.1) | 7 (20.6) | |
| **Tumor location** | | | 0.262 | | | 0.523 |
| Jejunum, *n* (%) | 2 (3.8) | 0 | | 1 (2.7) | 2 (5.7) | |
| Ileum, *n* (%) | 50 (96.2) | 32 (100) | | 36 (97.3) | 33 (94.3) | |
| **Somatostatin analogue treatment** | | | 0.582 | | | 0.951 |
| Yes, *n* (%) | 31 (59.6) | 21 (65.6) | | 23 (62.2) | 22 (62.9) | |
| **Chemotherapy** | | | 0.180 | | | 0.994 |
| No, *n* (%) | 42 (80.8) | 30 (93.8) | | 31 (83.8) | 29 (82.9) | |
| Preoperative, *n* (%) | 4 (7.7) | 0 | | 1 (2.7) | 1 (2.9) | |
| Postoperative, *n* (%) | 6 (11.5) | 2 (6.3) | | 5 (13.5) | 5 (14.3) | |
| **Multiple primary tumors** | | | 0.319 | | | 0.664 |
| Yes, *n* (%) | 16 (32.0) | 7 (21.9) | | 12 (33.3) | 10 (28.6) | |
| **P-CgA** | | | | | | |
| Median (IQR) nmol/L | 6.30 (3.8250–20.25) | 6.70 (3.70–17.00) | 0.905 | 7.05 (3.75–12.00) | 4.80 (3.375–11.00) | 0.594 |
| **dU-5-HIAA** | | | | | | |
| Median (IQR) μmol/L | 77.00 (30.00–214.50) | 35.00 (24.00–254.50) | 0.275 | 84.00 (28.75–155.25) | 57.50 (40.00–176.25) | 0.950 |

P-CgA, plasma chromogranin A; dU-5-HIAA, 24 h urine 5-hydroxyindoleacetic acid; Classes T-, N- and M refer to Tumor, Node and Metastasis of the TNM-staging system.

intensity group (72.2% vs. 50.0%, p = 0.048) (Table 8). Median urine 5-hydroxyindoleacetic acid (5-HIAA) levels were significantly higher in SB-NETs with high cytoplasmic TLR9 intensity compared with low intensity (122 μmol/L vs. 43 μmol/L, p = 0.004) (Table 9).

**Table 3. Baseline characteristics and comparison stratified by Toll-like receptor (TLR) 4 cytoplasmic intensity in primary tumors and lymph node metastases.**

| Variables | TLR4 primary tumor cytoplasmic intensity low, *n = 67* | TLR4 primary tumor cytoplasmic intensity high, *n = 49* | P between groups | TLR4 metastasis cytoplasmic intensity low, *n = 48* | TLR4 metastasis cytoplasmic intensity high, *n = 29* | P between groups |
|---|---|---|---|---|---|---|
| **Sex** | | | 0.127 | | | 0.616 |
| Male, *n* (%) | 41 (61.2) | 23 (46.9) | | 26 (54.2) | 14 (48.3) | |
| Female, *n* (%) | 26 (38.8) | 26 (53.1) | | 22 (45.8) | 15 (51.7) | |
| **Age, median (IQR) years** | 58.10 (50.10–72.17) | 68.01 (63.35–73.75) | **0.002** | 65.92 (55.51–71.36) | 66.51 (57.36–74.44) | 0.690 |
| **T-Class** | | | 0.167 | | | 0.900 |
| T1–2, *n* (%) | 21 (31.3) | 8 (16.3) | | 10 (20.8) | 7 (24.1) | |
| T3, *n* (%) | 26 (38.8) | 25 (51.0) | | 24 (50.0) | 13 (44.8) | |
| T4, *n* (%) | 20 (29.9) | 16 (32.7) | | 14 (29.2) | 9 (31.0) | |
| **N-Class** | | | 0.200 | | | |
| N0, *n* (%) | 16 (23.9) | 7 (14.3) | | | | |
| N1–2, *n* (%) | 51 (76.1) | 42 (85.7) | | | | |
| **M-Class** | | | 0.192 | | | 0.213 |
| M0, *n* (%) | 40 (59.7) | 35 (71.4) | | 28 (58.3) | 21 (72.4) | |
| M1, *n* (%) | 27 (40.3) | 14 (28.6) | | 20 (41.7) | 8 (27.6) | |
| **Stage** | | | 0.134 | | | 0.213 |
| I–II, *n* (%) | 10 (14.9) | 4 (8.2) | | 0 | 0 | |
| III, *n* (%) | 30 (44.8) | 31 (63.3) | | 28 (58.3) | 21 (72.4) | |
| IV, *n* (%) | 27 (40.3) | 14 (28.6) | | 20 (41.7) | 8 (27.6) | |
| **Grade** | | | 0.401 | | | 0.408 |
| 1, *n* (%) | 53 (81.5) | 36 (75.0) | | 36 (76.6) | 19 (67.9) | |
| 2, *n* (%) | 12 (18.5) | 12 (25.0) | | 11 (23.4) | 9 (31.2) | |
| **Tumor location** | | | 0.973 | | | 0.875 |
| Jejunum, *n* (%) | 4 (6.0) | 3 (6.1) | | 2 (4.2) | 1 (3.4) | |
| Ileum, *n* (%) | 63 (94.0) | 46 (93.9) | | 46 (95.8) | 28 (96.6) | |
| **Somatostatin analogue treatment** | | | 0.127 | | | 0.601 |
| Yes, *n* (%) | 41 (61.2) | 23 (46.9) | | 31 (64.6) | 17 (58.6) | |
| **Chemotherapy** | | | 0.131 | | | 0.167 |
| No, *n* (%) | 55 (82.1) | 44 (89.8) | | 41 (85.4) | 24 (82.8) | |
| Preoperative, *n* (%) | 2 (3.0) | 3 (6.1) | | 0 | 2 (6.9) | |
| Postoperative, *n* (%) | 10 (14.9) | 2 (4.1) | | 7 (14.6) | 3 (10.3) | |
| **Multiple primary tumors** | | | 0.324 | | | 0.403 |
| Yes, *n* (%) | 17 (26.2) | 17 (34.7) | | 12 (25.5) | 10 (34.5) | |
| **P-CgA** | | | | | | |
| Median (IQR) nmol/L | 5.90 (3.525–13.50) | 6.90 (3.80–15.50) | 0.485 | 6.10 (3.55–11.45) | 6.30 (3.45–14.25) | 0.827 |
| **dU-5-HIAA** | | | | | | |
| Median (IQR) μmol/L | 58.00 (27.00–156.00) | 48.50 (22.75–196.00) | 0.697 | 63 (35.75–131.25) | 105.50 (31.00–258.75) | 0.339 |

P-CgA, plasma chromogranin A; dU-5-HIAA, 24 h urine 5-hydroxyindoleacetic acid; Classes T-, N- and M refer to Tumor, Node and Metastasis of the TNM-staging system.

## Toll-like receptors and survival

Disease-specific survival in the whole cohort at 5 years was 89%, at 10 years 75.6% and at the end of follow up 57.3%.

**Table 4. Baseline characteristics and comparison stratified by Toll-like receptor (TLR) 5 cytoplasmic intensity in primary tumors and lymph node metastases.**

| Variables | TLR5 primary tumor cytoplasmic intensity low, *n* = 53 | TLR5 primary tumor cytoplasmic intensity high, *n* = 46 | P between groups | TLR5 metastasis cytoplasmic intensity low, *n* = 49 | TLR5 metastasis cytoplasmic intensity high, *n* = 9 | P between groups |
|---|---|---|---|---|---|---|
| **Sex** | | | 0.411 | | | 0.451 |
| Male, *n* (%) | 32 (60.4) | 24 (52.2) | | 26 (53.1) | 6 (66.7) | |
| Female, n (%) | 21 (39.6) | 22 (47.8) | | 23 (46.9) | 3 (33.3) | |
| **Age, median (IQR) years** | 61.34 (51.75–71.30) | 65.84 (55.67–73.05) | 0.351 | 64.83 (56.00–72.56) | 67.99 (55.10–74.36) | 0.660 |
| **T-Class** | | | 0.974 | | | 0.703 |
| T1–2, *n* (%) | 14 (26.4) | 13 (28.3) | | 11 (22.4) | 3 (33.3) | |
| T3, *n* (%) | 22 (41.5) | 19 (41.3) | | 23 (46.9) | 3 (33.3) | |
| T4, *n* (%) | 17 (32.1) | 14 (30.4) | | 15 (30.6) | 3 (33.3) | |
| **N-Class** | | | 0.148 | | | |
| N0, *n* (%) | 13 (24.5) | 6 (13.0) | | | | |
| N1–2, *n* (%) | 40 (75.5) | 40 (87.0) | | | | |
| **M-Class** | | | 0.620 | | | 0.576 |
| M0, *n* (%) | 32 (60.4) | 30 (65.2) | | 32 (65.3) | 5 (55.6) | |
| M1, *n* (%) | 21 (39.6) | 16 (34.8) | | 17 (34.7) | 4 (44.4) | |
| **Stage** | | | 0.638 | | | 0.576 |
| I–II, *n* (%) | 8 (15.1) | 5 (10.9) | | 0 | 0 | |
| III, *n* (%) | 24 (45.3) | 25 (54.3) | | 32 (65.3) | 5 (55.6) | |
| IV, *n* (%) | 21 (39.6) | 16 (34.8) | | 17 (34.7) | 4 (44.4) | |
| **Grade** | | | 0.484 | | | 0.602 |
| 1, *n* (%) | 41 (78.8) | 32 (72.7) | | 36 (75.0) | 6 (66.7) | |
| 2, *n* (%) | 11 (21.2) | 12 (27.3) | | 12 (25.0) | 3 (33.3) | |
| **Tumor location** | | | 0.533 | | | 0.446 |
| Jejunum, *n* (%) | 2 (3.8) | 3 (6.5) | | 3 (6.1) | 0 | |
| Ileum, *n* (%) | 51 (96.2) | 43 (93.5) | | 46 (93.9) | 9 (100) | |
| **Somatostatin analogue treatment** | | | 0.425 | | | 0.520 |
| Yes, *n* (%) | 33 (62.3) | 25 (54.3) | | 27 (55.1) | 6 (66.7) | |
| **Chemotherapy** | | | 0.874 | | | 0.062 |
| No, *n* (%) | 45 (84.9) | 39 (84.8) | | 44 (89.8) | 7 (77.8) | |
| Preoperative, *n* (%) | 2 (3.8) | 1 (2.2) | | 0 | 1 (11.1) | |
| Postoperative, *n* (%) | 6 (11.3) | 6 (13.0) | | 5 (10.2) | 1 (11.1) | |
| **Multiple primary tumors** | | | 0.650 | | | 0.761 |
| Yes, *n* (%) | 14 (26.9) | 14 (31.1) | | 13 (27.1) | 2 (22.2) | |
| **P-CgA** | | | | | | |
| Median (IQR) nmol/L | 6.80 (3.525–19.00) | 5.50 (3.60–9.90) | 0.510 | 5.70 (3.30–9.85) | 5.95 (4.425–13.10) | 0.534 |
| **dU-5-HIAA** | | | | | | |
| Median (IQR) µmol/L | 42.00 (24.00–153.00) | 92.00 (37.25–198.750) | 0.258 | 56.00 (27.00–105.00) | 67.50 (44.25–331.75) | 0.266 |

P-CgA, plasma chromogranin A; dU-5-HIAA, 24 h urine 5-hydroxyindoleacetic acid; Classes T-, N- and M refer to Tumor, Node and Metastasis of the TNM-staging system.

None of the primary tumor TLR intensities associated with DSS. Although high cytoplasmic TLR8 intensity in primary tumors seemed to associate with worse survival compared with low intensity group (49.5% vs. 72.5%, p = 0.068), but the difference was not statistically significant. In lymph node metastasis, high TLR7 intensity was associated with significantly

**Table 5. Baseline characteristics and comparison stratified by Toll-like receptor (TLR) 5 nucleic intensity in primary tumors and lymph node metastases.**

| Variables | TLR5 primary tumor nucleic intensity low, *n = 62* | TLR5 primary tumor nucleic intensity high, *n = 37* | P between groups | TLR5 metastasis nucleic intensity low, *n = 45* | TLR5 metastasis nucleic intensity high, *n = 13* | P between groups |
|---|---|---|---|---|---|---|
| **Sex** | | | 0.976 | | | 0.458 |
| Male, *n* (%) | 35 (56.5) | 21 (56.8) | | 26 (57.8) | 6 (46.2) | |
| Female, n (%) | 27 (43.5) | 16 (43.2) | | 19 (42.2) | 7 (53.8) | |
| **Age, median (IQR) years** | 61.68 (51.71–72.19) | 66.05 (56.19–73.38) | 0.272 | 64.33 (55.39–71.53) | 67.99 (62.33–82.63) | 0.095 |
| **T-Class** | | | 0.726 | | | 0.658 |
| T1–2, *n* (%) | 17 (27.4) | 10 (27.0) | | 12 (26.7) | 2 (15.4) | |
| T3, *n* (%) | 24 (38.7) | 17 (45.9) | | 20 (44.4) | 6 (46.2) | |
| T4, *n* (%) | 21 (33.9) | 10 (27.0) | | 13 (28.9) | 5 (38.5) | |
| **N-Class** | | | 0.268 | | | |
| N0, *n* (%) | 14 (22.6) | 5 (13.5) | | | | |
| N1–2, *n* (%) | 48 (77.4) | 32 (86.5) | | | | |
| **M-Class** | | | 0.432 | | | 0.263 |
| M0, *n* (%) | 37 (59.7) | 25 (67.6) | | 27 (60.0) | 10 (76.9) | |
| M1, *n* (%) | 25 (40.3) | 12 (32.4) | | 18 (40.0) | 3 (23.1) | |
| **Stage** | | | 0.533 | | | 0.263 |
| I–II, *n* (%) | 9 (14.5) | 4 (10.8) | | 0 | 0 | |
| III, *n* (%) | 28 (45.2) | 21 (56.8) | | 27 (60.0) | 10 (76.9) | |
| IV, *n* (%) | 25 (40.3) | 12 (32.4) | | 18 (40.0) | 3 (23.1) | |
| **Grade** | | | 0.422 | | | 0.678 |
| 1, *n* (%) | 48 (78.7) | 25 (71.4) | | 33 (75.0) | 9 (69.2) | |
| 2, *n* (%) | 13 (21.3) | 10 (28.6) | | 11 (25.0) | 4 (30.8) | |
| **Tumor location** | | | **0.043** | | | 0.339 |
| Jejunum, *n* (%) | 1 (1.6) | 4 (10.8) | | 3 (6.7) | 0 | |
| Ileum, *n* (%) | 61 (98.4) | 33 (89.2) | | 42 (93.3) | 13 (100) | |
| **Somatostatin analogue treatment** | | | 0.775 | | | **0.005** |
| Yes, *n* (%) | 37 (59.7) | 21 (56.8) | | 30 (66.7) | 3 (23.1) | |
| **Chemotherapy** | | | 0.524 | | | 0.073 |
| No, *n* (%) | 54 (87.1) | 30 (81.1) | | 39 (86.7) | 12 (92.3) | |
| Preoperative, *n* (%) | 1 (1.6) | 2 (5.4) | | 0 | 1 (7.7) | |
| Postoperative, *n* (%) | 7 (11.3) | 5 (13.5) | | 6 (13.3) | 0 | |
| **Multiple primary tumors** | | | 0.778 | | | 0.308 |
| Yes, *n* (%) | 17 (27.9) | 11 (30.6) | | 13 (29.5) | 2 (15.4) | |
| **P-CgA** | | | | | | |
| Median (IQR) nmol/L | 6.4 (3.75–15.50) | 4.90 (3.025–9.95) | 0.439 | 4.80 (3.350–11.00) | 6.70 (3.10–8.90) | 0.867 |
| **dU-5-HIAA** | | | | | | |
| Median (IQR) µmol/L | 42.00 (25.00–153.0) | 107.00 (37.250–203.50) | 0.230 | 63.00 (30.250–139.00) | 51.00 (28.00–77.00) | 0.462 |

P-CgA, plasma chromogranin A; dU-5-HIAA, 24 h urine 5-hydroxyindoleacetic acid; Classes T-, N- and M refer to Tumor, Node and Metastasis of the TNM-staging system.

worse DSS compared to low intensity (26.4% vs. 84.9%, p = 0.028) (Fig 2). Survival percentages are presented in Table 11.

Crude and adjusted HRs for disease specific mortality are presented in Table 12. Adjusted HR in high cytoplasmic TLR7 lymph node metastasis intensity group was 3.90 (95% CI 1.07–14.3) when compared to low intensity.

**Table 6. Baseline characteristics and comparison stratified by Toll-like receptor (TLR) 6 cytoplasmic intensity in primary tumors and lymph node metastases.**

| Variables | TLR6 primary tumor cytoplasmic intensity low, *n = 74* | TLR6 primary tumor cytoplasmic intensity high, *n = 41* | P between groups | TLR6 metastasis cytoplasmic intensity low, *n = 53* | TLR6 metastasis cytoplasmic intensity high, *n = 23* | P between groups |
|---|---|---|---|---|---|---|
| **Sex** | | | 0.135 | | | 0.481 |
| Male, *n* (%) | 45 (60.8) | 19 (46.3) | | 30 (56.6) | 11 (47.8) | |
| Female, n (%) | 29 (39.2) | 22 (53.7) | | 23 (43.4) | 12 (52.2) | |
| **Age, median (IQR)** years | 65.55 (54.26–73.13) | 65.23 (57.64–73.08) | 0.898 | 66.51 (55.54–71.85) | 65.23 (56.27–69.90) | 0.591 |
| **T-Class** | | | 0.261 | | | 0.097 |
| T1–2, *n* (%) | 23 (31.1) | 7 (17.1) | | 15 (28.3) | 2 (8.7) | |
| T3, *n* (%) | 30 (40.5) | 20 (48.8) | | 25 (47.2) | 11 (47.8) | |
| T4, *n* (%) | 21 (28.4) | 14 (34.1) | | 13 (24.5) | 10 (43.5) | |
| **N-Class** | | | 0.790 | | | |
| N0, *n* (%) | 16 (21.6) | 8 (19.5) | | | | |
| N1–2, *n* (%) | 58 (78.4) | 33 (80.5) | | | | |
| **M-Class** | | | 0.511 | | | 0.908 |
| M0, *n* (%) | 46 (62.2) | 28 (68.3) | | 33 (62.3) | 14 (60.9) | |
| M1, *n* (%) | 28 (37.8) | 13 (31.7) | | 20 (37.7) | 9 (39.1) | |
| **Stage** | | | 0.741 | | | 0.908 |
| I–II, *n* (%) | 10 (13.5) | 5 (12.2) | | 0 | 0 | |
| III, *n* (%) | 36 (48.6) | 23 (56.1) | | 33 (62.3) | 14 (60.9) | |
| IV, *n* (%) | 28 (37.8) | 13 (35.7) | | 20 (37.7) | 9 (39.1) | |
| **Grade** | | | 0.784 | | | 0.903 |
| 1, *n* (%) | 57 (79.2) | 30 (76.9) | | 37 (72.5) | 17 (73.9) | |
| 2, *n* (%) | 15 (20.8) | 9 (23.1) | | 14 (27.5) | 6 (26.1) | |
| **Tumor location** | | | 0.221 | | | 0.906 |
| Jejunum, *n* (%) | 3 (4.1) | 4 (9.8) | | 2 (3.8) | 1 (4.3) | |
| Ileum, *n* (%) | 71 (95.9) | 37 (90.2) | | 51 (96.2) | 22 (95.7) | |
| **Somatostatin analogue treatment** | | | 0.059 | | | 0.288 |
| Yes, *n* (%) | 46 (62.2) | 18 (43.9) | | 30 (56.6) | 16 (69.6) | |
| **Chemotherapy** | | | 0.486 | | | 0.638 |
| No, *n* (%) | 63 (85.1) | 34 (82.9) | | 44 (83.0) | 20 (87.0) | |
| Preoperative, *n* (%) | 2 (2.7) | 3 (7.3) | | 1 (1.9) | 1 (4.3) | |
| Postoperative, *n* (%) | 9 (12.2) | 4 (9.8) | | 8 (15.1) | 2 (8.7) | |
| **Multiple primary tumors** | | | 0.890 | | | 0.174 |
| Yes, *n* (%) | 21 (28.8) | 12 (30.0) | | 17 (32.7) | 4 (17.4) | |
| **P-CgA** | | | | | | |
| Median (IQR) nmol/L | 6.40 (3.90–15.50) | 5.80 (3.20–14.50) | 0.524 | 6.50 (3.90–14.00) | 6.40 (3.10–13.50) | 0.499 |
| **dU-5-HIAA** | | | | | | |
| Median (IQR) μmol/L | 46.00 (23.50–198.50) | 92.00 (37.00–133.00) | 0.312 | 76.50 (35.75–196.00) | 76.00 (44.00–199.00) | 0.808 |

P-CgA, plasma chromogranin A; dU-5-HIAA, 24 h urine 5-hydroxyindoleacetic acid; Classes T-, N- and M refer to Tumor, Node and Metastasis of the TNM-staging system.

## Discussion

In the present study, we discovered TLR 1, 2, 4, 5, 6, 7, 8, and 9 expression in SB-NETs and lymph node metastases. High TLR7 intensity in lymph node metastases associated with worse

**Table 7. Baseline characteristics and comparison stratified by Toll-like receptor (TLR) 7 cytoplasmic intensity in primary tumors and lymph node metastases.**

| Variables | TLR7 primary tumor cytoplasmic intensity low, *n = 74* | TLR7 primary tumor cytoplasmic intensity high, *n = 37* | P between groups | TLR7 metastasis cytoplasmic intensity low, *n = 46* | TLR7 metastasis cytoplasmic intensity high, *n = 26* | P between groups |
|---|---|---|---|---|---|---|
| **Sex** | | | 0.892 | | | 0.594 |
| Male, *n* (%) | 41 (55.4) | 21 (56.8) | | 26 (56.5) | 13 (50.0) | |
| Female, *n* (%) | 33 (44.6) | 16 (43.2) | | 20 (43.5) | 13 (50.0) | |
| **Age, median (IQR)** years | 65.35 (52.04–73.74) | 65.23 (58.83–69.46) | 0.798 | 67.03 (55.41–72.29) | 65.80 (57.82–74.30) | 0.622 |
| **T-Class** | | | 0.741 | | | **0.049** |
| T1–2, *n* (%) | 21 (28.4) | 8 (21.6) | | 15 (32.6) | 2 (7.7) | |
| T3, *n* (%) | 30 (40.5) | 16 (43.2) | | 19 (41.3) | 13 (50.0) | |
| T4, *n* (%) | 23 (31.1) | 13 (35.1) | | 12 (26.1) | 11 (42.3) | |
| **N-Class** | | | 0.239 | | | |
| N0, *n* (%) | 17 (23.0) | 5 (13.5) | | | | |
| N1–2, *n* (%) | 57 (77.0) | 32 (86.5) | | | | |
| **M-Class** | | | 0.576 | | | 0.899 |
| M0, *n* (%) | 46 (62.2) | 25 (67.6) | | 29 (63.0) | 16 (61.5) | |
| M1, *n* (%) | 28 (37.8) | 12 (32.4) | | 17 (37.0) | 10 (38.5) | |
| **Stage** | | | 0.505 | | | 0.899 |
| I–II, *n* (%) | 10 (13.5) | 3 (8.1) | | 0 | 0 | |
| III, *n* (%) | 36 (48.6) | 22 (59.5) | | 29 (63.0) | 16 (61.5) | |
| IV, *n* (%) | 28 (37.8) | 12 (32.4) | | 17 (37.0) | 10 (38.5) | |
| **Grade** | | | 0.947 | | | |
| 1, *n* (%) | 58 (79.5) | 28 (80.0) | | 33 (75.0) | 19 (73.1) | 0.859 |
| 2, *n* (%) | 15 (20.5) | 7 (20.0) | | 11 (25.0) | 7 (26.9) | |
| **Tumor location** | | | **0.027** | | | 0.183 |
| Jejunum, *n* (%) | 2 (2.7) | 5 (13.5) | | 3 (6.5) | 0 | |
| Ileum, *n* (%) | 72 (93.7) | 32 (86.5) | | 43 (93.5) | 26 (100) | |
| **Somatostatin analogue treatment** | | | 0.280 | | | 0.655 |
| Yes, *n* (%) | 44 (59.5) | 18 (48.6) | | 29 (63.0) | 15 (57.7) | |
| **Chemotherapy** | | | **0.048** | | | 0.733 |
| No, *n* (%) | 64 (86.5) | 31 (83.8) | | 39 (84.8) | 23 (88.5) | |
| Preoperative, *n* (%) | 1 (1.4) | 4 (10.8) | | 1 (2.2) | 1 (3.8) | |
| Postoperative, *n* (%) | 9 (12.2 = | 2 (5.4) | | 6 (13.0) | 2 (7.7) | |
| **Multiple primary tumors** | | | 0.967 | | | 0.594 |
| Yes, *n* (%) | 20 (27.4) | 10 (27.8) | | 13 (28.9) | 6 (23.1) | |
| **P-CgA** | | | | | | |
| Median (IQR) nmol/L | 6.3 (3.75–15.75) | 6.60 (3.10–12.00) | 0.689 | 6.40 (4.00–10.00) | 3.95 (3.10–18.75) | 0.440 |
| **dU-5-HIAA** | | | | | | |
| Median (IQR) µmol/L | 46.00 (23.50–144.50) | 95.50 (37.75–221.50) | 0.144 | 76.00 (40.00–169.00) | 67.00 (28.00–339.50) | 0.841 |

P-CgA, plasma chromogranin A; dU-5-HIAA, 24 h urine 5-hydroxyindoleacetic acid; Classes T-, N- and M refer to Tumor, Node and Metastasis of the TNM-staging system.

survival both in univariate model and after adjusting for confounders. SB-NETs with high TLR8 expression had worse survival compared to low expression, but the difference was not statistically significant. TLRs 1, 2, 4, 5, 6, and 9 showed no association with survival.

**Table 8. Baseline characteristics and comparison stratified by Toll-like receptor (TLR) 8 cytoplasmic intensity in primary tumors and lymph node metastases.**

| Variables | TLR8 primary tumor cytoplasmic intensity low, *n = 79* | TLR8 primary tumor cytoplasmic intensity high, *n = 34* | P between groups | TLR8 metastasis cytoplasmic intensity low, *n = 40* | TLR8 metastasis cytoplasmic intensity high, *n = 36* | P between groups |
|---|---|---|---|---|---|---|
| **Sex** | | | 0.102 | | | 0.663 |
| Male, *n* (%) | 48 (60.8) | 15 (44.1) | | 22 (55.0) | 18 (50.0) | |
| Female, n (%) | 31 (39.2) | 19 (55.9) | | 18 (45.0) | 18 (50.0) | |
| **Age, median (IQR) years** | 65.23 (54.20–72.19) | 67.36 (56.69–73.44) | 0.499 | 68.75 (55.51–75.04) | 65.43 (57.14–69.38) | 0.344 |
| **T-Class** | | | 0.867 | | | 0.119 |
| T1–2, *n* (%) | 20 (25.3) | 10 (29.4) | | 10 (25.0) | 6 (16.7) | |
| T3, *n* (%) | 34 (43.0) | 13 (38.2) | | 22 (55.0) | 15 (41.7) | |
| T4, *n* (%) | 25 (31.6) | 11 (32.4) | | 8 (20.0) | 15 (41.7) | |
| **N-Class** | | | 0.218 | | | |
| N0, *n* (%) | 13 (16.5) | 9 (26.5) | | | | |
| N1–2, *n* (%) | 66 (83.5) | 25 (73.5) | | | | |
| **M-Class** | | | 0.563 | | | 0.727 |
| M0, *n* (%) | 51 (64.6) | 20 (58.8) | | 24 (60.0) | 23 (63.9) | |
| M1, *n* (%) | 28 (35.4) | 14 (41.2) | | 16 (40.0) | 13 (36.1) | |
| **Stage** | | | 0.772 | | | 0.727 |
| I–II, *n* (%) | 8 (10.1) | 4 (11.8) | | 0 | 0 | |
| III, *n* (%) | 43 (54.4) | 16 (47.1) | | 24 (60.0) | 23 (63.9) | |
| IV, *n* (%) | 28 (35.4) | 14 (41.2) | | 16 (40.0) | 13 (36.1) | |
| **Grade** | | | 0.993 | | | 0.921 |
| 1, *n* (%) | 61 (78.2) | 25 (78.1) | | 29 (72.5) | 25 (73.5) | |
| 2, *n* (%) | 17 (21.8) | 7 (21.9) | | 11 (27.5) | 9 (26.5) | |
| **Tumor location** | | | 0.859 | | | 0.619 |
| Jejunum, *n* (%) | 4 (5.1) | 2 (5.9) | | 2 (5.0) | 1 (2.8) | |
| Ileum, *n* (%) | 75 (94.9) | 32 (94.1) | | 38 (95.0) | 35 (97.2) | |
| **Somatostatin analogue treatment** | | | 0.758 | | | **0.048** |
| Yes, *n* (%) | 44 (55.7) | 20 (58.8) | | 20 (50.0) | 26 (72.2) | |
| **Chemotherapy** | | | 0.892 | | | 0.881 |
| No, *n* (%) | 67 (84.8) | 30 (88.2) | | 33 (82.5) | 31 (86.1) | |
| Preoperative, *n* (%) | 3 (3.8) | 1 (2.9) | | 1 (2.5) | 1 (2.8) | |
| Postoperative, *n* (%) | 9 (11.4) | 3 (8.8) | | 6 (15.0) | 4 (11.1) | |
| **Multiple primary tumors** | | | 0.824 | | | 0.776 |
| Yes, *n* (%) | 22 (28.2) | 10 (30.3) | | 12 (30.8) | 10 (27.8) | |
| **P-CgA** | | | | | | |
| Median (IQR) nmol/L | 6.35 (3.475–16.00) | 7.00 (3.975–20.25) | 0.379 | 5.80 (3.650–12.00) | 6.30 (3.30–15.00) | 0.839 |
| **dU-5-HIAA** | | | | | | |
| Median (IQR) µmol/L | 76.00 (28.00–198.00) | 43.00 (21.50–180.750) | 0.350 | 78.00 (38.00–198.00) | 57.00 (31.50–187.50) | 0.711 |

P-CgA, plasma chromogranin A; dU-5-HIAA, 24 h urine 5-hydroxyindoleacetic acid; Classes T-, N- and M refer to Tumor, Node and Metastasis of the TNM-staging system.

This study has several strengths. The present study was to our knowledge the first study to characterize the expression of TLRs in NETs overall, and specifically in SB-NETs. Our study included both samples from primary tumors and lymph node metastases for extended understanding of the subject in question. Our cohort included over a hundred patients, which is a

**Table 9. Baseline characteristics and comparison stratified by Toll-like receptor (TLR) 9 cytoplasmic intensity in primary tumors and lymph node metastases.**

| Variables | TLR9 primary tumor cytoplasmic intensity low, *n* = 89 | TLR9 primary tumor cytoplasmic intensity high, *n* = 22 | P between groups | TLR9 metastasis cytoplasmic intensity low, *n* = 54 | TLR9 metastasis cytoplasmic intensity high, *n* = 16 | P between groups |
|---|---|---|---|---|---|---|
| **Sex** | | | 0.412 | | | 0.857 |
| Male, *n* (%) | 48 (53.9) | 14 (63.6) | | 29 (53.7) | 9 (56.3) | |
| Female, *n* (%) | 41 (46.1) | 8 (36.4) | | 25 (46.3) | 7 (43.8) | |
| **Age, median (IQR) years** | 61.34 (53.43–71.52) | 69.46 (65.84–73.75) | **0.012** | 65.55 (56.13–70.54) | 58.83 (45.01–74.48) | 0.467 |
| **T-Class** | | | 0.470 | | | 0.192 |
| T1–2, *n* (%) | 24 (27.0) | 4 (18.2) | | 10 (18.5) | 5 (31.3) | |
| T3, *n* (%) | 36 (40.4) | 12 (54.5) | | 25 (46.3) | 9 (56.3) | |
| T4, *n* (%) | 29 (32.6) | 6 (27.3) | | 19 (35.2) | 2 (12.5) | |
| **N-Class** | | | 0.224 | | | |
| N0, *n* (%) | 18 (20.2) | 2 (9.1) | | | | |
| N1–2, *n* (%) | 71 (79.8) | 20 (90.9) | | | | |
| **M-Class** | | | 0.950 | | | 0.252 |
| M0, *n* (%) | 56 (62.9) | 14 (63.6) | | 32 (59.3) | 12 (75.0) | |
| M1, *n* (%) | 33 (37.1) | 8 (36.4) | | 22 (40.7) | 4 (25.0) | |
| **Stage** | | | 0.537 | | | 0.252 |
| I–II, *n* (%) | 11 (12.4) | 1 (4.5) | | 0 | 0 | |
| III, *n* (%) | 45 (50.6) | 13 (59.1) | | 32 (59.3) | 12 (75.0) | |
| IV, *n* (%) | 33 (37.1) | 8 (36.4) | | 22 (40.7) | 4 (25.0) | |
| **Grade** | | | 0.436 | | | 0.187 |
| 1, *n* (%) | 69 (79.3) | 15 (71.4) | | 41 (78.8) | 10 (62.5) | |
| 2, *n* (%) | 18 (20.7) | 6 (28.6) | | 11 (21.2) | 6 (37.5) | |
| **Tumor location** | | | 0.247 | | | 0.659 |
| Jejunum, *n* (%) | 3 (3.4) | 2 (9.1) | | 2 (3.7) | 1 (6.2) | |
| Ileum, *n* (%) | 86 (96.6) | 20 (90.9) | | 52 (96.3) | 15 (93.8) | |
| **Somatostatin analogue treatment** | | | 0.475 | | | 0.865 |
| Yes, *n* (%) | 52 (58.4) | 11 (50.0) | | 35 (64.8) | 10 (62.5) | |
| **Chemotherapy** | | | 0.930 | | | 0.446 |
| No, *n* (%) | 76 (85.4) | 19 (86.4) | | 47 (87.0) | 13 (81.3) | |
| Preoperative, *n* (%) | 3 (3.4) | 1 (4.5) | | 2 (3.7) | 0 | |
| Postoperative, *n* (%) | 10 (11.2) | 2 (9.1) | | 5 (9.3) | 3 (18.8) | |
| **Multiple primary tumors** | | | 0.445 | | | 0.936 |
| Yes, *n* (%) | 27 (31.0) | 5 (22.7) | | 16 (30.2) | 5 (31.3) | |
| **P-CgA** | | | | | | |
| Median (IQR) nmol/L | 6.30 (3.60–14.00) | 4.70 (3.00–36.00) | 0.702 | 6.30 (3.60–14.00) | 5.00 (3.00–14.00) | 0.527 |
| **dU-5-HIAA** | | | | | | |
| Median (IQR) μmol/L | 43.00 (25.00–164.25) | 122.00 (92.00–269.50) | **0.004** | 68.00 (40.00–198.00) | 112.00 (25.00–187.50) | 0.839 |

P-CgA, plasma chromogranin A; dU-5-HIAA, 24 h urine 5-hydroxyindoleacetic acid; Classes T-, N- and M refer to Tumor, Node and Metastasis of the TNM-staging system.

large cohort in the field of NET research due to the slowly progressing nature and rarity of the disease. Many large studies today are done by tissue microarray techniques, but we were able to use whole section slides including selection of deepest invasion area. Our study has an extensive follow-up period which is of utmost importance when investigating diseases such as

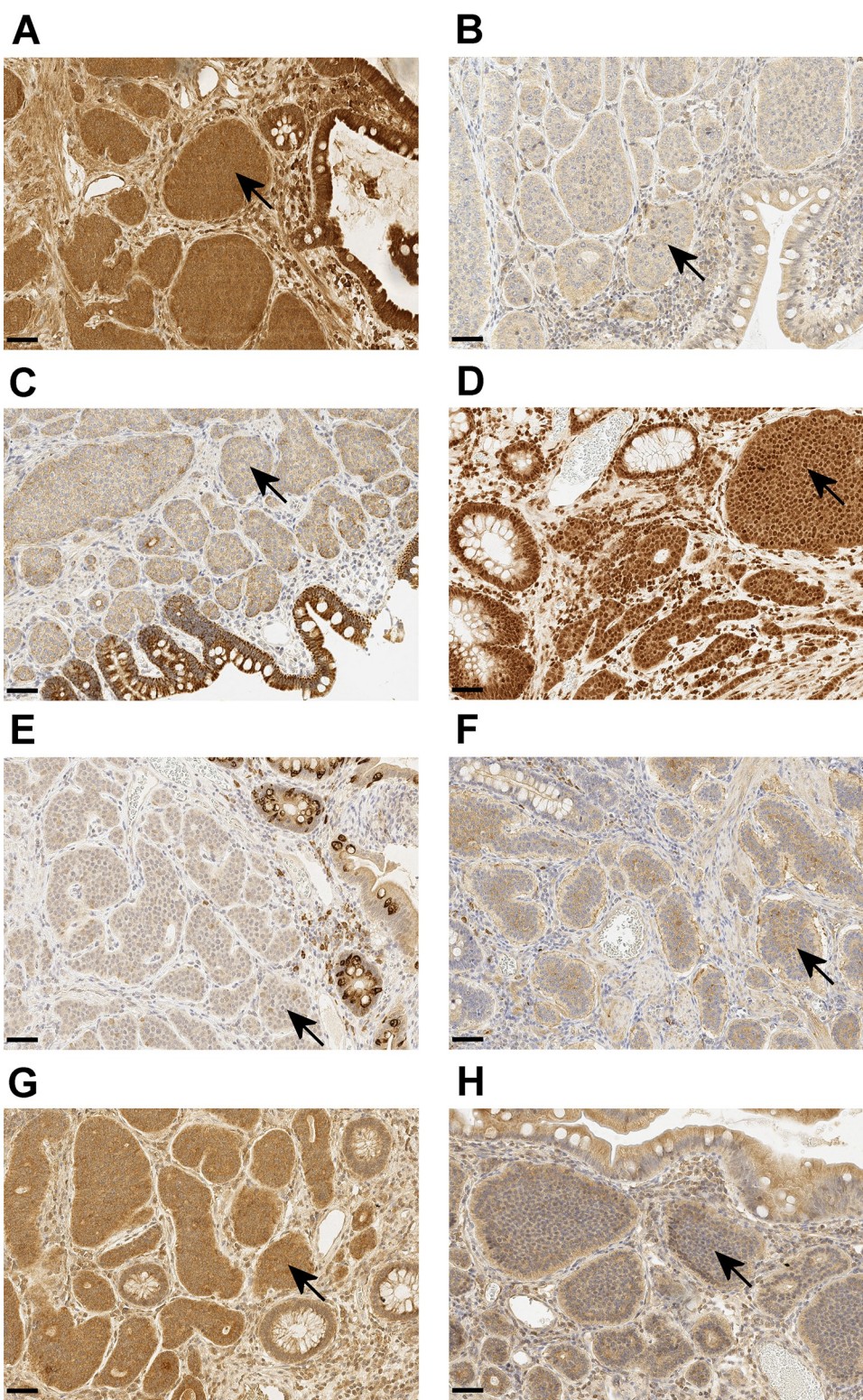

**Fig 1. Staining examples.** Immunohistochemical staining examples of Toll-like receptors (TLRs) 1, 2, 4, 5, 6, 7, 8 and 9 in representative small bowel neuroendocrine tumor samples showing (**A**) High TLR1 cytoplasmic intensity, (**B**) intermediate TLR2 cytoplasmic intensity, (**C**) intermediate TLR4 cytoplasmic intensity, (**D**) high TLR5 cytoplasmic and nucleic intensity, (**E**) Intermediate TLR6 cytoplasmic intensity, (**F**) intermediate TLR7 cytoplasmic intensity, (**G**) high TLR8 cytoplasmic intensity and (**H**) intermediate TLR9 cytoplasmic intensity in x20 magnification. The scale bar length is 50 μm (bottom left corner). Arrows indicate TLR-positive tumor cell islets.

**Table 10. Median Toll-like receptor (TLR) staining intensity in primary small bowel neuroendocrine tumors and lymph node metastases.**

|  | Primary tumors | Lymph node metastases |
|---|---|---|
| **TLR1** |  |  |
| Cytoplasmic | 2 (IQR 1.5–2) | 2 (IQR 1.5–2) |
| **TLR2** |  |  |
| Cytoplasmic | 1.5 (IQR 1–2) | 1.5 (IQR 1.5–2) |
| **TLR4** |  |  |
| Cytoplasmic | 1.5 (IQR 1–2) | 2 (IQR 1.5–2.5) |
| **TLR5** |  |  |
| Cytoplasmic | 1.5 (IQR 1–2.5) | 2.5 (IQR 2–2.5) |
| Nucleic | 1.5 (IQR 0–2.5) | 2.5 (IQR 2–2.5) |
| **TLR6** |  |  |
| Cytoplasmic | 1 (IQR 1–1.5) | 1 (1–1.5) |
| **TLR7** |  |  |
| Cytoplasmic | 1.5 (IQR 1–2) | 2 (IQR 1.5–2.5) |
| **TLR8** |  |  |
| Cytoplasmic | 1.5 (IQR 1–2) | 1 (IQR 1–2) |
| **TLR9** |  |  |
| Cytoplasmic | 2 (IQR 1–2). | 2 (IQR 1.5–2). |

IQR, interquartile range.

SB-NETs which have relatively slow disease progression. Finland is known for meticulous statistics, and we were provided with reliable and complete registry data by Statistics Finland considering both causes and times of death. This allowed for accurate DSS analyses. Our study also has limitations. The study population consisted only of G1 and G2 SB-NETs so the results cannot be applied to G3 SB-NETs or small bowel neuroendocrine carcinomas (SB-NECs). This is often the case with NET research due to the rarity and typically inoperable nature of G3/NEC disease, yet it leaves a gap in research knowledge when it comes to the most aggressive forms of this disease. The number of disease-specific deaths can be argued to be small, even though the follow-up period was extensive. The relatively small size of the cohort and small number of deaths create uncertainty to the results and replication studies are needed for more robust conclusions. The study is also limited by the different numbers of available samples per TLR. Some samples stained inconsistently and were excluded from the analyses, while others did not have adequate tissue material for all stainings. The hospitals also later limited the availability of new samples to preserve the diagnostic samples. Also since the study is exploratory, multiple statistical testing induces a risk of chance findings and confirmatory studies are deemed necessary.

Previous NET studies have investigated TLRs mostly in Merkel cell carcinoma (MCC), an aggressive, rare neuroendocrine skin malignancy often caused by Merkel cell polyomavirus (MCV). A study by Jouhi et al characterized TLR 2, 4, 5, 7 and 9 expressions in 128 patients with MCC [28]. They found preliminary association with TLR expression and clinicopathological variables, such as MCV status (positive/negative), tumor size and patient age. Like their findings, we also found high TLR4 expression to correlate with older age. In 2023, Imon et al used computational formulation to create a multiepitope candidate vaccine against MCV, incorporating TLR4 agonist as an adjuvant for highest immune response activation [29]. The candidate vaccine was tested in silico, where it seemed to promote immune memory development and natural immune protection against MCV. A pilot study by Bhatia et al investigated

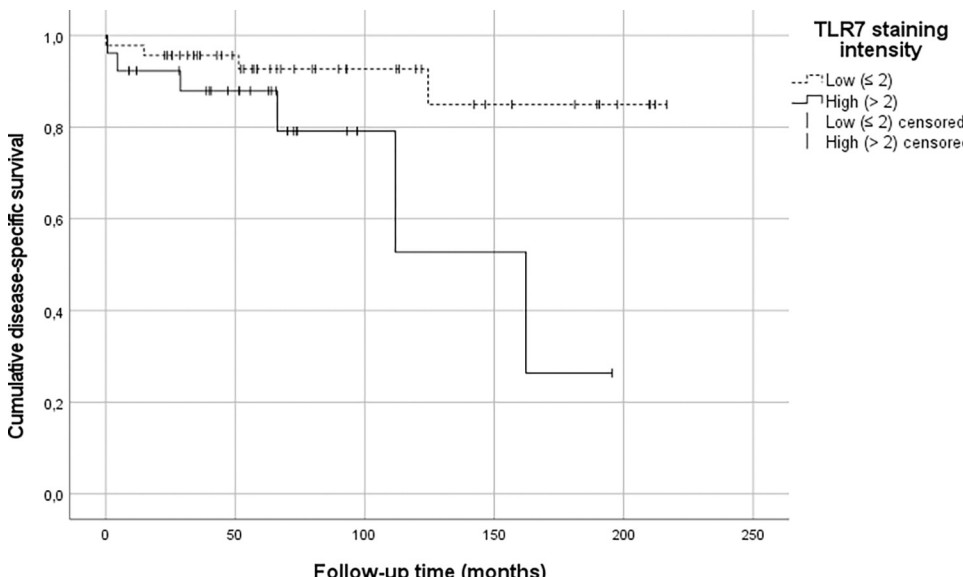

**Fig 2. Kaplan-Meier curve.** Cumulative disease-specific survival (DSS) in small bowel neuroendocrine tumor lymph node metastasis stratified by Toll like-receptor (TLR) 7 cytoplasmic staining intensity. The Kaplan-Meier curves for DSS differentiate between low and high TLR7 expression (p = 0.028).

intratumoral injections of G100, a TLR4 agonist, in patients with MCC [30]. The patients showed activation of pro-inflammatory changes, which associated with tumor regression. Sample size in the study was small and they found no correlation between clinical responses and baseline TLR expression. A study by Young et al analyzed 600 immune-related genes in pancreatic neuroendocrine tumors (PanNET), among them being TLR3 [31]. In Young's study, high TLR3 gene expression in a specific subtype of PanNET called metastasis-like primary (MLP)-1 was associated with poorer overall survival and the study also suggests that TLR3 may affect immune related gene expression and immune escape in MLP-1. The study also showed TLR7 gene enrichment in MLP-1 [31]. No further literature about TLRs in NETs could be found.

The role of TLRs in other cancers is controversial. It seems that the anti- or protumor effects of TLRs depend on the TLR is question, the cancer type and the immune cells infiltrating the tumor [23]. A recent 2023 study by Çelik et al on breast cancer found high TLR9 expression to associate with invasive carcinoma and low expression to associate with increased overall survival [32]. TLR7 and TLR8 activation in non-small cell lung cancer induced survival of cancer cell lines and increased cancer cell chemoresistance [33]. In gastric cancer TLR4 and TLR5 activation induced cancer cell survival [34]. In esophageal cancer TLR3 was associated with invasion and lymph node metastasis, TLR4 with lymph node metastasis and TLR7 with worse histological grade [35]. In colorectal cancer, TLR2 expression may contribute to sporadic carcinogenesis and TLR4 may increase cancer cell survival [34]. In murine models, TLR 7/8 agonist R848 strengthened the antitumor effect of both oxaliplatin in colorectal cancer and of stereotactic body radiotherapy in pancreatic ductal adenocarcinoma [36,37]. While TLR 7/8 agonist seems to boost the antitumor effect of different treatments, according to a meta-analysis, high TLR7 expression in tumors seems to lead to worse survival in a number of cancers [38]. TLR7 agonists can also stimulate these TLR7-expressing tumor cells [39].

TLR4 expression has been associated with advanced stage and poor prognosis in esophageal adenocarcinoma. Also, TLR 1/2/6 -network was shown to be upregulated in Barrett's

**Table 11. Disease-specific survival rates based on Toll-like receptor (TLR) 1, 2, 4, 5, 6, 7, 8, and 9 cytoplasmic staining intensity and TLR5 nucleic staining intensity in both primary tumors and lymph node metastases.**

| Disease-specific survival | No. of patients | TLR1 cytoplasmic intensity, low | TLR1 cytoplasmic intensity, high | p |
|---|---|---|---|---|
| Primary tumors | 116 | 67.2% | 54.1% | 0.518 |
| Lymph node metastases | 70 | 70.5% | 33.3% | 0.236 |
| | No. of patients | TLR2 cytoplasmic intensity, low | TLR2 cytoplasmic intensity, high | p |
| Primary tumors | 84 | 58.0% | 68.1% | 0.975 |
| Lymph node metastases | 72 | 79.5% | 70.4% | 0.343 |
| | No. of patients | TLR4 cytoplasmic intensity low | TLR4 cytoplasmic intensity, high | p |
| Primary tumors | 116 | 64.2% | 67.3% | 0.956 |
| Lymph node metastases | 77 | 68.5% | 63.3 | 0.329 |
| | No. of patients | TLR5 cytoplasmic intensity low | TLR5 cytoplasmic intensity, high | p |
| Primary tumors | 99 | 64.4% | 65.3% | 0.497 |
| Lymph node metastases | 58 | 67.3% | 100% | 0.363 |
| | No. of patients | TLR5 nucleic intensity low | TLR5 nucleic intensity, high | p |
| Primary tumors | 99 | 64.2% | 64.8% | 0.457 |
| Lymph node metastases | 58 | 65.7% | 84.6% | 0.508 |
| | No. of patients | TLR6 cytoplasmic intensity low | TLR6 cytoplasmic intensity, high | p |
| Primary tumors | 115 | 65.6% | 69.4% | 0.701 |
| Lymph node metastases | 76 | 68.7% | 71.7% | 0.803 |
| | No. of patients | TLR7 cytoplasmic intensity low | TLR7 cytoplasmic intensity, high | p |
| Primary tumors | 111 | 64.1% | 75.7% | 0.278 |
| Lymph node metastases | 72 | 84.9% | 26.4% | **0.028** |
| | No. of patients | TLR8 cytoplasmic intensity low | TLR8 cytoplasmic intensity, high | p |
| Primary tumors | 113 | 72.5% | 49.5% | 0.068 |
| Lymph node metastases | 76 | 69.3% | 65.0% | 0.727 |
| | No. of patients | TLR9 cytoplasmic intensity low | TLR9 cytoplasmic intensity, high | p |
| Primary tumors | 111 | 62.4% | 71.4% | 0.612 |
| Lymph node metastases | 70 | 62.6% | 87.5% | 0.788 |

metaplasia, esophageal dysplasia and esophageal cancer. [40]. The significance of luminal pathogen sensing TLRs in the esophagus may be related to the passage and direct contact of microbial material with luminal epithelial cells. The cancer cells in SB-NETs are not directly in contact with the bowel lumen. Although some bacteria may pass through the mucosa, we speculate SB-NETs to have less contact with luminal TLR ligands and more contact with endogenous ligands. This could be a factor affecting the TLR expression and signaling in SB-NETs and a possible reason why TLRs 1, 2, 4 and 6 did not seem to have prognostic significance in

**Table 12. Hazard ratios (HR) for disease-specific mortality with 95% confidence intervals (CI) in small bowel neuroendocrine tumor lymph node metastasis stratified by Toll-like receptor (TLR) 7 staining intensity (low/high).**

| | TLR7 intensity | | | |
|---|---|---|---|---|
| Lymph node metastasis | No. of patients | Low HR (95%CI) | High HR (95%CI) | |
| Crude | 72 | 1.00 (reference) | 3.90 (1.07–14.3) | |
| Adjusted | 72 | 1.00 (reference) | 7.45 (1.25–44.5) | |

Adjusted for age (continuous), sex (male/female), stage (I-II, III, IV), grade of differentiation (G1 or G2) and somatostatin therapy (no/yes).

SB-NETs. Also, intrinsic ligand sensing TLRs could be hypothesized to have clinical relevance in non-luminal tumors, such as SB-NETs. In the present study, TLR7, recognizing intrinsic ligands, seemed to be prognostic in lymph node metastases, which are also not in direct contact of the bowel lumen. In esophageal adenocarcinoma, intrinsic factor sensing TLRs 3, 7 and 8 seemed to be upregulated in early events of esophageal carcinogenesis but had no association with prognosis, highlighting the tumor specific differences in TLR expression and clinical significance [41].

The present study has potential clinical implications. TLR7 agonists and antagonists are available and could potentially be used in treatment of SB-NETs by either strengthening the antitumor effects of other treatment modalities or by direct antitumor effect. The study did not investigate TLR function, and because almost all TLRs were expressed in SB-NETs, TLRs could be tested as drug therapy targets despite most of them showing no association with survival in the present study.

## Conclusions

TLRs 1–2 and 4–9 are expressed in SB-NETs and SB-NET lymph node metastases. High TLR7 expression in lymph node metastases associates with worse disease specific survival. Due to large cohort size, the use of whole section slides and complete registry data, the results are strong, yet preliminary. To determine the future direction in this field of research, confirmatory studies with even larger cohorts and inclusion of G3/NEC disease are warranted–especially regarding the observed survival association.

## Supporting information

**S1 Dataset. Anonymized dataset of the statistical analyses.** The data is in.sav format. IBM SPSS Statistics program is required to handle the data.
(XLSX)

## Acknowledgments

Special thanks: Erja Tomperi, Tanja Kuusisto, Riitta Vuento, Sanna Mari Koskela, Harri Kaikkonen, Tuomas Moilanen, Tarja Piispanen and Biobank Borealis of Northern Finland for their excellent technical assistance.

## Author Contributions

**Conceptualization:** Niko Hiltunen, Olli Helminen.

**Formal analysis:** Niko Hiltunen, Olli Helminen.

**Funding acquisition:** Niko Hiltunen, Olli Helminen.

**Investigation:** Niko Hiltunen, Niko Kemi, Olli Helminen.

**Project administration:** Niko Hiltunen, Olli Helminen.

**Resources:** Jan Böhm.

**Supervision:** Olli Helminen.

**Visualization:** Niko Hiltunen.

**Writing – original draft:** Niko Hiltunen, Olli Helminen.

**Writing – review & editing:** Niko Hiltunen, Niko Kemi, Juha P. Väyrynen, Jan Böhm, Joonas H. Kauppila, Heikki Huhta, Olli Helminen.

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
