## [Decision Letter · Decision Letter 0]

20 Dec 2023

PONE-D-23-35599Toll-like receptors 1–9 in small bowel neuroendocrine tumors – Clinical significance and prognosisPLOS ONE

Dear Dr. Hiltunen,

Thank you for submitting your manuscript to PLOS ONE. After careful consideration, we feel that it has merit but does not fully meet PLOS ONE’s publication criteria as it currently stands. Therefore, we invite you to submit a revised version of the manuscript that addresses the points raised during the review process.

**After careful evaluation of the reviewer’s comments, it has been decided that your manuscript requires major revisions before it can be considered for publication. I want to emphasize that this choice is not a reflection of the caliber of your work overall; rather, it is a measure to make sure the manuscript satisfies the high standards of excellence that our journal demands.**

We look forward to receiving your revised manuscript.

Kind regards,

Mohsan Ullah

Academic Editor

PLOS ONE

Journal Requirements:

Additional Editor Comments:

Following the careful consideration of reviewers comments, I encourage authors to revise the reviewers concerns and submit the revised manuscript.

Reviewers' comments:

Reviewer's Responses to Questions

**Comments to the Author**

1. Is the manuscript technically sound, and do the data support the conclusions?

Reviewer #1: No

Reviewer #2: Yes

2. Has the statistical analysis been performed appropriately and rigorously? 

Reviewer #1: Yes

Reviewer #2: Yes

3. Have the authors made all data underlying the findings in their manuscript fully available?

Reviewer #1: Yes

Reviewer #2: No

4. Is the manuscript presented in an intelligible fashion and written in standard English?

Reviewer #1: Yes

Reviewer #2: Yes

5. Review Comments to the Author

Reviewer #1: Dear authors

The study has significant scientific importance and has interesting findings. Methods and sample size is suitable and sufficient. The discussion section can be improved using recent studies from 2022 and 2023. I is acceptable in the present form.

Kind regards

Reviewer #2: The manuscript (PONE-D-23-35599) entitled Toll-like receptors 1–9 in small bowel neuroendocrine tumors – Clinical significance and prognosis' has been carefully reviewed as per the guidelines of the journal. In my point of view, the manuscript is judged to be of potential interest to the readers and the scientific community as well. While the overall concept of the paper is reasonably satisfactory, it falls within the scope of the journal. However, it has some corrections. The manuscript still has a number of flaws, which need to be addressed before it is formally accepted for publication. For instance, there are some errors in the manuscript's scientific, typographical, and grammatical content. Overall, this manuscript may offer important contributions to the literature, but needs some minor corrections for possible publication in this journal

6. PLOS authors have the option to publish the peer review history of their article (what does this mean?). If published, this will include your full peer review and any attached files.

Reviewer #1: No

Reviewer #2: No

---

## [Author Response · Author response to Decision Letter 0]

1 Feb 2024

Responses are listed in a separate file labelled "Response to Reviewers".

---

## [Decision Letter · Decision Letter 1]

15 Apr 2024

Toll-like receptors 1–9 in small bowel neuroendocrine tumors – Clinical significance and prognosis

PONE-D-23-35599R1

Dear Dr. Hiltunen,

We’re pleased to inform you that your manuscript has been judged scientifically suitable for publication and will be formally accepted for publication once it meets all outstanding technical requirements.

Kind regards,

Luwen Zhang

Academic Editor

PLOS ONE

Additional Editor Comments (optional):

Reviewers' comments:

Reviewer's Responses to Questions

**Comments to the Author**

1. If the authors have adequately addressed your comments raised in a previous round of review and you feel that this manuscript is now acceptable for publication, you may indicate that here to bypass the “Comments to the Author” section, enter your conflict of interest statement in the “Confidential to Editor” section, and submit your "Accept" recommendation.

Reviewer #1: (No Response)

Reviewer #2: All comments have been addressed

2. Is the manuscript technically sound, and do the data support the conclusions?

Reviewer #1: (No Response)

Reviewer #2: Yes

3. Has the statistical analysis been performed appropriately and rigorously? 

Reviewer #1: (No Response)

Reviewer #2: Yes

4. Have the authors made all data underlying the findings in their manuscript fully available?

Reviewer #1: (No Response)

Reviewer #2: Yes

5. Is the manuscript presented in an intelligible fashion and written in standard English?

Reviewer #1: (No Response)

Reviewer #2: Yes

6. Review Comments to the Author

Reviewer #1: (No Response)

Reviewer #2: (No Response)

7. PLOS authors have the option to publish the peer review history of their article (what does this mean?). If published, this will include your full peer review and any attached files.

Reviewer #1: No

Reviewer #2: No

---

## [Editor Report · Acceptance letter]

26 Apr 2024

PONE-D-23-35599R1 

PLOS ONE

Dear Dr. Hiltunen, 

I'm pleased to inform you that your manuscript has been deemed suitable for publication in PLOS ONE. Congratulations! Your manuscript is now being handed over to our production team.

Kind regards, 

on behalf of

Dr Luwen Zhang 

Academic Editor

PLOS ONE